



# Evaluating two methods of estimating error variances from multiple data sets using an error model

Therese Rieckh[1,2] and Richard Anthes[1]

[1]COSMIC Program Office, University Corporation for Atmospheric Research, Boulder, Colorado, U.S.A
[2]Wegener Center for Climate and Global Change, University of Graz, Graz, Austria

*Correspondence to*: Richard Anthes (anthes@ucar.edu)

**Abstract.** In this paper we compare two different methods of estimating the error variances of two or more independent data sets. One method, called the "three-cornered hat" (3CH) or "triple co-location" method, requires three data sets. Another method, which we call the "two-cornered hat" (2CH) method, requires only two data sets. Both methods assume that the errors

of the data sets are not correlated and are unbiased. The 3CH method has been used in previous studies to estimate the error variances associated with a number of physical and geophysical data sets. Braun et al. (2001) used the two-cornered hat (2CH) method to estimate the error variances associated with two observational data sets of total atmospheric water vapor.

In this paper we compare the 3CH and 2CH methods using a simple error model to simulate three and two data sets with

various error correlations and biases. With this error model, we know the exact error variances and covariances, which we use to assess the accuracy of the 3CH and 2CH estimates. We examine the sensitivity of the estimated error variances to the degree of error correlation between two of the data sets as well as the sample size. We find that the 3CH method is less sensitive to these factors than the 2CH method and hence is more accurate. We also find that biases in one of the data sets has a minimal effect on the 3CH method, but can produce large errors in the 2CH method.

## 1 Introduction

In atmospheric sciences, observations and models are often combined with the goal of providing accurate and complete representations of the current or future state of the atmosphere. Knowing the error characteristics of observations and models is important to understanding the degree to which atmospheric phenomena of interest are accurately described and analyzed. Estimating observational and modelling error characteristics are thus of inherent scientific interest. In addition, knowing the

error characteristics are important for practical applications such as data assimilation and numerical weather prediction. In many modern data assimilation schemes, observations of a given type are weighted proportionally to the inverse of their error variance (e.g. Desroziers and Ivanov, 2001).





There are at least three somewhat similar methods for estimating the error variances associated with two or more data sets. The "three-cornered hat" (3CH) or "triple co-location method" has been used in physics, oceanography and other scientific disciplines to estimate the errors associated with three independent data sets. Braun et al. (2001) combined two independent data sets, Global Positioning System (GPS) slant water vapor (SWV) and water vapor radiometer (WVR), to estimate the SWV

and WVR errors. In analogy to the 3CH method, we refer to the Braun et al. (2001) method as the two-cornered hat (2CH) method. Kuo et al. (2004) and Chen et al. (2011) used the "apparent error method," which is a variation of the 2CH method, to estimate the error of radio occultation (RO) observations using the known (or estimated) error variance of a forecast model.

The 3CH method was originally developed as the "N-cornered hat" method (Gray and Allan, 1974) for estimating error

variances from $N$ atomic clocks. Requiring three data sets at a minimum, the 3CH method produces estimates of the error variances of each of three data sets. W.J. Wriley (2003) provides a summary of the 3CH method and its history (http://www.wriley.com/3-CornHat.htm). Variations and enhancements of the method have been applied to many diverse geophysical data sets. The 3HC method has been used to estimate the stability of GNSS clocks using the measured frequencies from multiple clocks (Ekstrom and Koppang, 2006; Griggs et al., 2014, 2015; Luna et al., 2017). Valty et al. (2013) used the

3CH method to estimate the geophysical load deformation computed from GRACE satellites, GPS vertical displacement measurements, and global general circulation (GCM) models. Anthes and Rieckh (2018) used the 3CH method to estimate the error variances of three observational (two radio occultation retrievals and radiosondes) and two model data sets using various combinations of the five data sets.

The 3CH method, termed the "triple-co-location method" by Stoffelen (1998) in his study of estimated ocean surface winds using three data sets, has been widely used in the field of oceanography and hydrometeorology (e.g. Su et al., 2014; Gruber et al. 2016). O'Carroll et al. (2008) compared three systems to measure sea-surface temperatures: two different radiometers and in situ observations from buoys. They discuss the assumption of neglecting the error correlations among the three data sets and the effect of representativeness errors. Roebeling et al. (2012) used the triple co-location method to estimate the errors

associated with three ways of estimating precipitation: the Spinning Enhanced Visible and Infrared Imager (SEVERI), weather radars, and ground-based rain gauges.

The major assumption in all of the above methods is that the errors of the three systems are uncorrelated and unbiased. Correlations between any or all of the three measurement systems will reduce the accuracy of the error estimates. Other factors

that can reduce the accuracy include widely different errors associated with the three systems or a small sample size. These factors can lead to negative estimates of error variances, especially when the estimates are close to zero.

In this paper we estimate the effect of neglecting the error covariances using two or three simulated data sets for which the true error variances and covariances are known. We develop an error model to simulate the data sets with random and bias



errors using a set of assumed True profiles. We then calculate the true error variance and covariance terms in the three simulated data sets and show the impact of neglecting these terms on the estimated error variances.

## 2 Error estimates using the 2CH and 3CH methods

We assume we have three data sets X, Y and Z that are all measuring the same physical variable, e.g. specific humidity, q, at

5    the same location and time.

The *error variance* of the data set X is defined as

$$\mathrm{VAR_{err}}(X) = (1/n) \sum (X - \mathrm{True})^2 = (1/n) \sum X_{err}^2 \tag{1}$$

where True is the true (but unknown) value of X (as well as Y and Z), $X_{err} = (X - \mathrm{True})$, and n is the number of samples. In general, the errors of X, Y, and Z may be correlated or not.

### 2.1 Three-cornered hat (3CH) method

In the 3CH method, the relationship between the mean square differences of X and Y, MS(X − Y), and their error variances

15    and covariances are given by Eqs. (7) – (9) of Anthes and Rieckh (2018)

$$\mathrm{VAR_{err}}(X) = \tfrac{1}{2}[\mathrm{MS}(X - Y) + \mathrm{MS}(X - Z) - \mathrm{MS}(Y - Z)]$$
$$+ \mathrm{COV_{err}}(X,Y) + \mathrm{COV_{err}}(X,Z) - \mathrm{COV_{err}}(Y,Z) \tag{2}$$

20   $$\mathrm{VAR_{err}}(Y) = \tfrac{1}{2}[\mathrm{MS}(X - Y) + \mathrm{MS}(Y - Z) - \mathrm{MS}(X - Z)]$$
$$+ \mathrm{COV_{err}}(X,Y) + \mathrm{COV_{err}}(Y,Z) - \mathrm{COV_{err}}(X,Z) \tag{3}$$

$$\mathrm{VAR_{err}}(Z) = \tfrac{1}{2}[\mathrm{MS}(X - Z) + \mathrm{MS}(Y - Z) - \mathrm{MS}(X - Y)]$$
$$+ \mathrm{COV_{err}}(X,Z) + \mathrm{COV_{err}}(Y,Z) - \mathrm{COV_{err}}(X,Y) \tag{4}$$

The last three covariance terms in Eqs. (2) – (4) are the terms that we neglect when using real data to estimate the error variances of X, Y and Z.





### 2.2 Two-cornered hat (2CH) method

In the 2CH method, there are only two data sets, X and Z. To derive the relationship between the error variances of X and Z

given the sums and differences of the two data sets, we first write

5    X = (True + $X_{err}$) and Z = (True + $Z_{err}$), then add X and Z and square the sum

$$(X + Z)^2 = 4True^2 + 4True(X_{err} + Z_{err}) + (X_{err} + Z_{err})^2 \tag{5}$$

Equation (5) is summed over all the data pairs to get

$$\Sigma(X + Z)^2 = 4\Sigma True^2 + \Sigma\{(X_{err} + Z_{err})^2 + 4True(X_{err} + Z_{err})\} \tag{6}$$

$$MS(X + Z) = 4MS(True) + VAR_{err}(X) + VAR_{err}(Z)$$
$$+ 2COV_{err}(X,Z) + 4M(True, X_{err} + Z_{err}) \tag{7}$$

where

$$M(X) = (1/n)\, \Sigma X$$
$$M(X,Y) = M(X*Y) = (1/n)\, \Sigma(X*Y)$$

We then subtract Z from X and square the difference to get
$$MS(X - Z) = VAR_{err}(X) + VAR_{err}(Z) - 2COV_{err}(X,Z) \tag{8}$$

Finally, we solve for MS(True) by subtracting Eq. (8) from Eq. (7)

$$4*MS(True) = MS(X+Z) - MS(X - Z) - 4COV_{err}(X,Z) - 4M(True, X_{err}+Z_{err}). \tag{9}$$

By squaring the expression X = (True + $X_{err}$) we get the exact expression for the $VAR_{err}(X)$

30    $$VAR_{err}(X) = MS(X) - MS(True) - 2M(True, X_{err}) \tag{10}$$

and substituting for MS(True) from Eq. (9) gives




$$VAR_{err}(X) = MS(X) - [MS(X + Z) - MS(X - Z)]/4$$
$$- 2M(True,X_{err}) + COV_{err}(X,Z) + M(True,X_{err}+Z_{err}). \tag{11}$$

Omitting the last three error terms, we obtain:

$$\mathbf{VAR_{err}(X)_{est} = MS(X) - [MS(X + Z) - MS(X - Z)]/4.} \tag{11a}$$

Similarly, we obtain

$$VAR_{err}(Z) = MS(Z) - [MS(X + Z) - MS(X - Z)]/4$$
$$- 2M(True,Z_{err}) + COV_{err}(X,Z) + M(True,X_{err}+Z_{err}) \tag{12}$$

and again neglecting the last three error terms we obtain

$$\mathbf{VAR_{err}(Z)_{est} = MS(Z) - [MS(X + Z) - MS(X - Z)]/4.} \tag{12a}$$

Equations (11a) and (12a) are equivalent to Eqs. (12) and (13) of Braun et al. (2011).

**2.3 Side note: apparent error method**

We note that Eq. (8) is the basis for the "apparent error" (AE) method in which X is an observed data set ($X_{obs}$) and Z is a
forecast of the X data set ($X_{fcst}$)

$$AE = X_{obs} - X_{fcst}$$

$$VAR_{err}(X_{obs}) = MS(X_{obs} - X_{fcst}) - VAR_{err}(X_{fcst}) + COV_{err}(X_{obs},X_{fcst})$$

The apparent error is equivalent to the Observation minus Background (O-B) statistic used in data assimilation studies. In the apparent error method, the correlation of errors between the observations and forecasts is assumed negligible and the error variance of the forecast is obtained from an independent estimate.

**2.4 Comparison of neglected terms in 3CH and 2CH methods**

In the 3CH method, the neglected error terms when computing $VAR_{err}(X)$ with Eq. (2) are





$COV_{err}(X,Y) + COV_{err}(X,Z) - COV_{err}(Y,Z).$

In the 2CH method, the neglected error terms when computing $VAR_{err}(X)$ with Eq. (11) are

$-2M(True,X_{err}) + COV_{err}(X,Z) + M(True,X_{err}+Z_{err}).$

We note that the neglected error terms in the 2CH method contain terms involving the product of True with errors, unlike in the 3CH method. Because True is typically an order of magnitude greater than the errors, these terms are likely much larger

than the neglected terms involving only products of errors, as in the 3CH method. We also note that if the X and Z errors are random and uncorrelated, all of the error terms will be zero for an infinite sample size. However, for finite sample sizes, these terms will be non-zero even if these conditions are met.

### 3 Generation of True data set and three simulated data sets with errors

We first generate a set of n vertical profiles of a variable, True, which we take as specific humidity from the ERA-Interim

reanalysis (Dee et al., 2011). We next generate three data sets X, Y and Z that are approximations of True (True plus errors), where the errors of X and Z are correlated to a degree that we can control. For simplicity, we assume the errors for Y are always uncorrelated with those of X and Z. This is analogous to a system of three observational systems in which the errors of two of them are correlated, but the errors of the third are not. We then look at the magnitude of the error terms in the 2CH and 3CH methods with various assumed correlation coefficients between the errors in X and Z and compare the estimated error

variances of X, Y and Z with their true error variances. Our tests will show the impact of the neglect of the error terms depending on the degree of error correlation between X and Z.

### 3.1 Model error profile

In Sections 3-6 we assume the mean error (bias) is zero for all the modeled data sets X, Y and Z. The effect of adding a bias error to Z is considered in Sect. 7. The assumed random error model for X is given by

$$X_{err} = random[-1.7,1.7]*CLIMO*STD \tag{13}$$

$$STD = 0.1 + 0.00042(1000-p) \tag{14}$$




where random[-1.7,1.7] is a random number between -1.7 and 1.7, CLIMO is the average of True over all n samples (called CLIMO in analogy to using a climatological value of $q$), and the standard deviation STD increases linearly with decreasing pressure p from a value of 0.1 (10%) at 1000 hPa to a value of 0.436 (43.6%) at 200 hPa. In the estimated error calculations below, we normalize $X_{err}$ by CLIMO so that it is expressed as a % error and the variance is expressed as $\%^2$.

## 3.2 Calculation of correlated errors

We first generate the random error profiles $X_{err}$, $Y_{err}$, and $Q_{err}$. All of these error profiles are uncorrelated. In general all three error profiles $X_{err}$, $Y_{err}$ and $Q_{err}$ may have different standard deviations, but for these tests we assume all standard deviations vary according to Eq. (14) for simplicity.

We now generate the error profile $Z_{err}$ as a linear combination of $X_{err}$ and $Q_{err}$:

$$Z_{err} = (aX_{err} + Q_{err})/(1 + a) \qquad (15)$$

where $a$ is a specified constant parameter that determines the degree of correlation between $Z_{err}$ and $X_{err}$. If $a=0$, $Z_{err}=Q_{err}$ and
the errors of X, Y and Z are all uncorrelated. If $a$ is not equal 0, the errors of Z are correlated with the errors of X. In this simple model, the errors of Z and Y and the errors of X and Y are always uncorrelated.

The correlation coefficient between the X and Z errors, $r_{xz}$ [also denoted by $r(X_{err},Z_{err})$], is given by

$r_{xz} = (1/n)\Sigma[X_{err} - (X_{err})_m][Z_{err} - (Z_{err})_m]/\sigma_x\sigma_z$ \qquad (16)

where $(X_{err})_m$ and $(Z_{err})_m$ are the mean values of the X and Z errors respectively (zero in our error model) and $\sigma_x$ and $\sigma_z$ are given by

$\sigma_x^2 = (1/n) \Sigma X_{err}^2$
$\sigma_z^2 = (1/n) \Sigma Z_{err}^2$

The correlation coefficient between $X_{err}$ and $Z_{err}$ may also be written as

$r(X_{err},Z_{err}) = COV_{err}(X,Z)/[\sigma(X_{err})\sigma(Z_{err})]$ where $\sigma(X_{err})$ and $\sigma(Z_{err})$ are the standard deviations of the X and Z errors respectively.





$$COV_{err}(X,Z) = r_{xz}\sigma_x\sigma_z \qquad (17)$$

where x and z are the normalized errors associated with X and Z respectively.

5  It can be shown that for this error model

$$r_{xz} = a/(1+a^2)^{1/2} \qquad (18a)$$

$$\sigma_z = [(1+a^2)^{1/2}/(1+a)]\,\sigma_x \qquad (18b)$$

$$\sigma_z^2 = (1+a^2)/(1+a)^2\,\sigma_x^2 \qquad (18c)$$

$$VAR_{err}(Z)=[(1+a^2)/(1+a)^2]VAR_{err}(X) \qquad (18d)$$

10  $$COV_{err}(X,Z) = [(a/(1+a)]VAR_{err}(X) \qquad (18e)$$

Thus the correlation between X and Z errors can be varied by varying the parameter $a$, as shown in Table 1. In the Table 1 example, the STD for the normalized $X_{err}$ is assumed to be a constant 10% (VAR = 100%$^2$) rather than varying the STD according to Eq. (14).

Table 1: Relationship between normalized error variances and standard deviations of data sets X and Z for different values of $a$. (x≡ normalized $X_{err}$ and z≡ normalized $Z_{err}$).

| $a$ | $r_{xz}$ | $\sigma_z/\sigma_x$ | $VAR_{err}(Z)(\%^2)$ | $COV_{err}(X,Z)\,(\%^2)$ |
|---|---|---|---|---|
| 0 | 0 | 1.0 | 1.00 $VAR_{err}(X)$ | 0.0 |
| 0.1 | 0.0995 | 0.9136 | 0.84 $VAR_{err}(X)$ | 0.0909 $VAR_{err}(X)$ |
| 0.3 | 0.287 | 0.803 | 0.65 $VAR_{err}(X)$ | 0.230 $VAR_{err}(X)$ |
| 0.5 | 0.447 | 0.745 | 0.55 $VAR_{err}(X)$ | 0.333 $VAR_{err}(X)$ |
| 1.0 | 0.707 | 0.707 | 0.50 $VAR_{err}(X)$ | 0.500 $VAR_{err}(X)$ |
| 2.0 | 0.894 | 0.745 | 0.55 $VAR_{err}(X)$ | 0.666 $VAR_{err}(X)$ |
| 10.0 | 0.995 | 0.913 | 0.83 $VAR_{err}(X)$ | 0.908 $VAR_{err}(X)$ |
| 100 | 0.99995 | 0.9901 | 0.98 $VAR_{err}(X)$ | 0.990 $VAR_{err}(X)$ |
| ∞ | 1.0 | 1.0 | 1.00 $VAR_{err}(X)$ | 1.00 $VAR_{err}(X)$ |

30  Note that $\sigma_z \leq \sigma_x$ and the maximum difference between $VAR_{err}(X)$ and $VAR_{err}(Z)$ occurs for $a=1$ when $\sigma_z$ is $0.707\sigma_x$ or $\sqrt{2}/2\,\sigma_x$.



### 3.3 Summary of generation of True data set and simulated data sets with errors

- We use 2007 ERA-Interim specific humidity $q$ profiles for a location near Minamidaitojima (hereafter Mina) Japan, which is located on Okinawa at 25.6°N 131.5°W. These are four model data profiles per day or n=1460 profiles.

- Assume each vertical profile of $q$ has no error. This is the True data set.

- Generate three different and independent random error profiles $X_{err}$, $Y_{err}$ and $Q_{err}$ from Eqs. (13) and (14).

- Generate $Z_{err}$ from Eq. (15) for various specified values of a.

- Add $X_{err}$, $Y_{err}$ and $Z_{err}$ to True to obtain the 1460 simulated profiles X, Y and Z respectively.

- Compute the estimated error variance profiles of X, Y and Z according to Eqs. (2), (3) and (4) neglecting the COV terms and compare with the true error variances, which can be computed exactly from the full Eqs. (2) – (4) including

  the known values of the covariance terms. All variance and covariance terms are normalized by the 2007 average value of specific humidity (CLIMO), computed from the True data set. A value of $a$=0 should give the most accurate estimation of error variances because all covariance terms will be close to zero (they won't be exactly zero because the sample size n is finite).

### 4 Example of simulated data profiles with random errors added

Figure 1a shows one set of simulated error profiles for X, Q, and several Z for different values of $a$.  The transition of $Z_{err}$ from $Q_{err}$ to $X_{err}$ can be seen easily in the black box around 600 hPa. $Q_{err}$ (bright green line) is positive (around 1 g kg$^{-1}$) and $Z_{err}$ for $a$=0 is identical to that. For $a$=0.1, $Z_{err}$ has a slightly smaller positive value, and $Z_{err}$ becomes negative as $a$ increases, becoming practically identical with $X_{err}$ (dotted line) for $a$=100.

Figure 1b shows the mean and standard deviation for the Z specific humidity profiles for $a$=0. Since the $Z_{err}$ are created by combining the two random errors $Q_{err}$ and $X_{err}$, the $Z_{err}$ are overall closer to zero (as can be seen in Fig. 1a in the black box.) This will result in a smaller standard deviation of Z if $0 < a < \infty$ (especially for values of $a$ close to 1), and will also decrease the true error variance of Z.



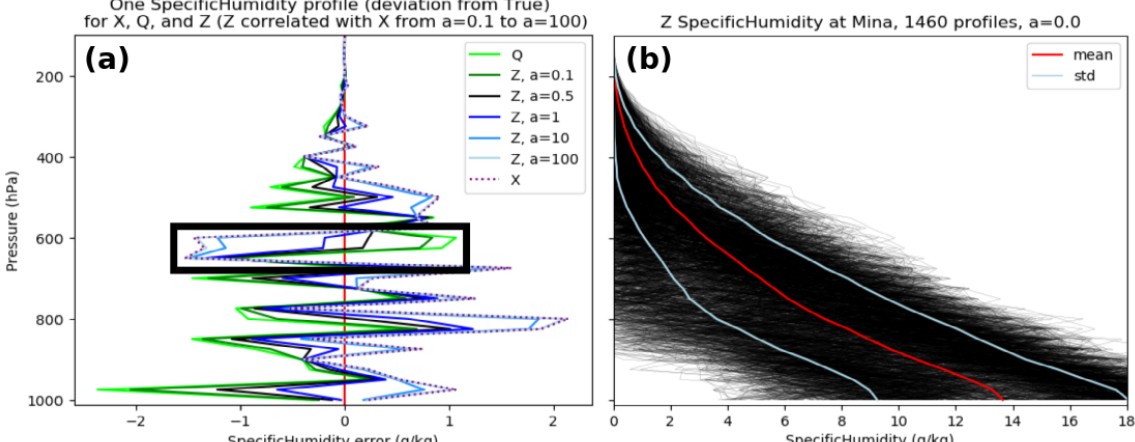

**Figure 1: (a) One set of error profiles with $X_{err}$ (dotted line), $Q_{err}$ (solid lime green line) and $Z_{err}$ (various solid lines for different values of $a$). $Q_{err}$ is uncorrelated with $X_{err}$. For $a=0$, $Z_{err}=Q_{err}$ and $Z_{err}$ and $X_{err}$ are uncorrelated. As $a$ increases, $Z_{err}$ is increasingly correlated with $X_{err}$ and looks less like $Q_{err}$ and more like $X_{err}$. For very large $a$ ($a=100$), $Z_{err}$ (light blue solid line) is almost equal to $X_{err}$ (dotted line). (b) 1460 profiles of data set Z with zero correlation with X ($a=0$).**

## 5 Effect of error correlations on estimated error variances

We now derive expressions for the estimated values of the error variances and their standard deviations for X, Y and Z for this error model and show how the correlations between $X_{err}$ and $Z_{err}$ affect the approximate values using the 3CH and 2CH methods. This will give some insight into how correlations between actual observed data sets will affect estimates of their error variances and standard deviations. To make results more readily comparable to previous studies, instead of showing the error variance, we show the square root of the error variance, or the error standard deviation, in most figures.

### 5.1 Effect of error correlations on 3CH method

In the 3CH method, three covariance terms are neglected. For uncorrelated errors between data sets, these terms are zero (for an infinite sample size). Correlation between the errors of two data sets will lead to a non-zero covariance term, which becomes larger for larger correlations. The error covariances $COV_{err}(X,Z)$, $COV_{err}(X,Y)$ and $COV_{err}(Y,Z)$ are shown in Fig. 2 for $a=0$ and 0.5. Note that the error covariances of X and Y (red line) and Y and Z (orange line) oscillate around zero (and are zero for





an infinitely large sample size) for $a=0$ (Fig. 2a). For $a = 0.5$, the $COV_{err}(X,Z)$ term increases with decreasing pressure, reaching a magnitude of about $800\%^2$ at 100 hPa (Fig. 2b).

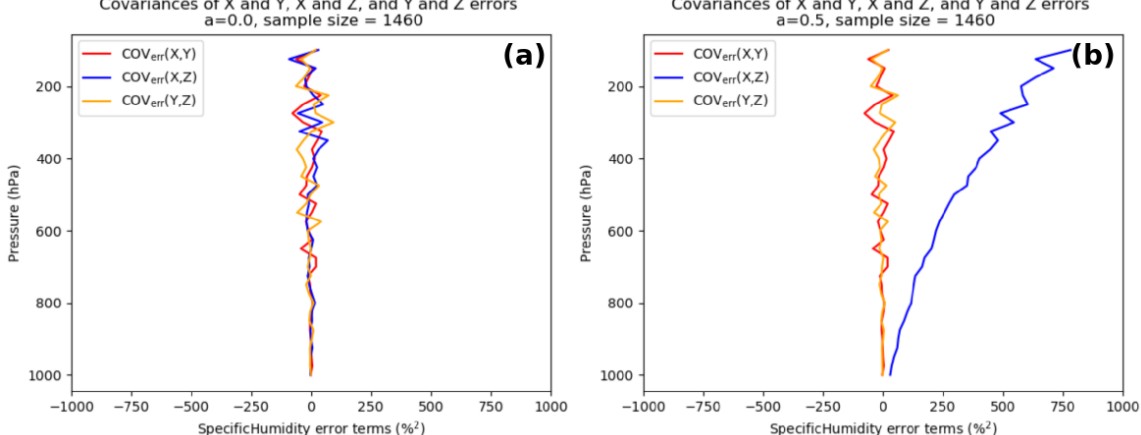

**Figure 2: Vertical profiles of normalized error covariances (X,Y), (Y,Z) and (X,Z) for: (a) $a=0$ and (b) $a= 0.5$.**

Completing the derivation of the error variances now, Eq. (2) becomes

$$VAR_{err}(X) = \tfrac{1}{2}[MS(X - Y) + MS(X - Z) - MS(Y - Z)]$$

$+ \cancel{COV_{err}(X,Y)} + COV_{err}(X,Z) - \cancel{COV_{err}(Y,Z)}$

where the terms crossed out are zero (for an infinite data set) because $Y_{err}$ is uncorrelated with $X_{err}$ and $Z_{err}$ in our error model.

In the three cornered hat method, the estimate of $VAR_{err}(X)$ is obtained by neglecting the term $COV_{err}(X,Z)$

$$VAR_{err}(X)_{est} = \tfrac{1}{2}[MS(X - Y) + MS(X - Z) - MS(Y - Z)] \qquad (2a)$$

or

$$VAR_{err}(X)_{est} = VAR_{err}(X) - COV_{err}(X,Z). \qquad (2b)$$

Using Eq. (18e) for this error model



$$\mathrm{VAR_{err}(X)_{est}} = [1/(1+a)]\ \mathrm{VAR_{err}(X)} \tag{2c}$$

Hence for $a>0$ the estimated error variance of X is always less than the true value, as seen in Fig. 3a.

We next consider the effect of the X and Z error correlation on the estimate for Y error variance. Crossing out the zero terms, Eq. (3) becomes

$$\mathrm{VAR_{err}(Y)} = \tfrac{1}{2}[\mathrm{MS(X-Y)} + \mathrm{MS(Y-Z)} - \mathrm{MS(X-Z)}]$$

$+\ \cancel{\mathrm{COV_{err}(X,Y)}} + \cancel{\mathrm{COV_{err}(Y,Z)})} - \mathrm{COV_{err}(X,Z)}$

or

$$\mathrm{VAR_{err}(Y)_{est}} = \mathrm{MS(X-Y)} + \mathrm{MS(Y-Z)} - \mathrm{MS(X-Z)} \tag{3a}$$

$$\mathrm{VAR_{err}(Y)_{est}} = \mathrm{VAR_{err}(Y)} + \mathrm{COV_{err}(X,Z)}. \tag{3b}$$

Substituting for the COV term from Eq. (18e) and noting that $\mathrm{VAR_{err}(X)} = \mathrm{VAR_{err}(Y)}$ we obtain

$\quad \mathrm{VAR_{err}(Y)_{est}} = [(1+2a)/(1+a)]\ \mathrm{VAR_{err}(Y)}. \tag{3c}$

Thus the estimated error variance for Y is always greater that the true value for $a>0$, which is seen in Fig. 3b.

Lastly, we consider the effect of the X and Z correlation on the estimate for the Z error variance. Eq. (4), with the zero terms
for this error model crossed out, becomes

$$\mathrm{VAR_{err}(Z)} = \tfrac{1}{2}[\mathrm{MS(X-Z)} + \mathrm{MS(Y-Z)} - \mathrm{MS(X-Y)}]$$

$$+ \mathrm{COV_{err}(X,Z)} + \cancel{\mathrm{COV_{err}(Y,Z)}} - \cancel{\mathrm{COV_{err}(X,Y)}}$$

or


$$\mathrm{VAR_{err}(Z)_{est}} = \mathrm{MS(X-Z)} + \mathrm{MS(Y-Z)} - \mathrm{MS(X-Y)} \tag{4a}$$

$$\mathrm{VAR_{err}(Z)_{est}} = \mathrm{VAR_{err}(Z)} - \mathrm{COV_{err}(X,Z)}. \tag{4b}$$





Substituting for the COV term from Eq. (18e) and using Eq. (18d) we obtain

$$\text{VAR}_{err}(Z)_{est} = [(1-a)/(1+a^2)] \, \text{VAR}_{err}(Z) \qquad\qquad (4d)$$

so that the estimated error variance for Z is less than the true value for $a>0$, which is illustrated in Fig. 3c. Finally, for $a>1$, Eq. (4d) shows that the estimated error variance of Z is negative and the STD is undefined. For $a=1.0$, the estimated error variance of Z is zero for an infinite data set, but oscillates around zero because of our finite data set. Thus the estimated STD of $Z_{err}$ is undefined at some levels (Fig. 3c).

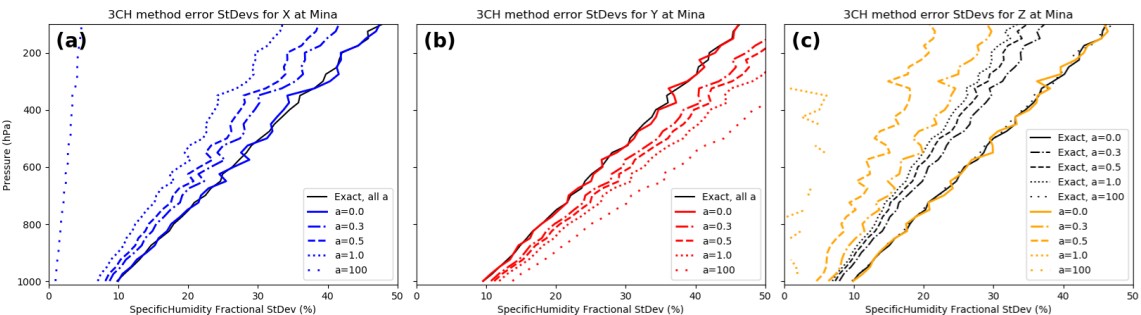

**Figure 3: (a) Estimated standard deviation of $X_{err}$ for values of $a$=0, 0.3, 0.5, 1.0 and 100 computed from Eq. (2a). Exact STD computed from data set X (solid black profile). (b) Same as (a) except for estimated STD of $Y_{err}$. (c) Same as (a) except for estimated STD of $Z_{err}$. Note that for different values of $a$, the exact STD of $X_{err}$ and $Y_{err}$ are always the same. The exact STD of $Z_{err}$ decreases as $a$ increases for $0<a<1.0$. For $a>1.0$, the exact STD of $Z_{err}$ increases as $a$ increases, becoming equal to the STD of $X_{err}$ for $a=\infty$ due to the way our error model is defined (see also Table 1). For $a$=100 the correlation between X and Z errors is 0.99995 (Table 1; black**
**dotted line for $a$=100 almost identical to black solid line for $a$=0).**

### 5.2 Summary of error correlations on 3CH method

The true values of STD($X_{err}$) and STD($Y_{err}$) are always the same, calculated from Eq. (14). The true values of STD($Z_{err}$) are similar to STD($X_{err}$) for $a$=0 and equal to STD($X_{err}$) for $a=\infty$. For $0<a<\infty$ STD($Z_{err}$) is less than STD($X_{err}$) and reaches a minimum of 0.707 STD($X_{err}$) for $a$=1 (Table 1).

The COV$_{err}$(X,Z), which is neglected in all of the approximate calculations, varies from 0 for $a$=0 to VAR$_{err}$(X) for large $a$. Thus for large $a$, from Eq. (2b) we find that the estimated VAR$_{err}$(X) tends to zero as $a$ increases.

For large $a$, from Eq. (3c) we see that the estimated value of VAR$_{err}$(Y) tends toward 2VAR$_{err}$(X). The approximate STD then
is therefore $\sqrt{2}$ times the true STD for large $a$, and this is seen in the plot for $a$=100 (Fig. 3b).





Figure 4 shows the ratios of the approximate error variances and standard deviations to the true values for $a$ ranging from 0 to 1 for the 3CH method. As the correlation parameter $a$ increases from 0, the ratios of the approximate to true errors increases. For a modest value of $a = 0.2$ (correlation coefficient between X and Z errors of 0.196), the errors in the STD estimates are −9 % for X, +8 % for Y and −12 % for Z. As the correlation between the X and Z error reach 0.5, the percentage errors for the X, Y and Z estimates reach −18 %, +14.5 % and −37 % respectively. To the extent that this error model gives an idea of the effect of the correlation between the errors of two of the three data sets, estimates of error standard deviations using a large sample of real data should be accurate within approximately 10% for correlation coefficients between data errors of 0.2 or below, and within 25% for correlation coefficients of around 0.3. The effect of the correlation between Z and X errors on the estimated error variance is greatest on the estimated Z error variance.

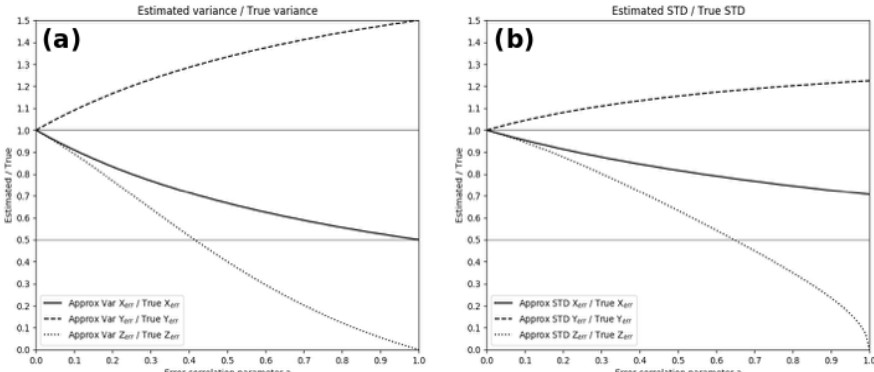

**Figure 4: Ratio of estimated (a) error variances to true variances and (b) estimated STD to true STD for the 3CH method for values of error correlation parameter $a$ ranging from 0 to 1.0.**

**5.3 Effect of error correlations on 2CH method**

We next examine how error correlations in our error model affect the 2CH method. To estimate the error variance of X using Eq. (11a), we omit the error terms $COV_{err}(X,Z)$, $M(True, X_{err})$, and $M(True, X_{err} + Z_{err})$. The $COV_{err}(X,Z)$ term was already shown in Fig. 2, and is the same in the 2CH method as in the 3CH method. Figures 5 and 6 show profiles of the other two terms for an error correlation between X and Z of $a=0$ and 0.5. In our error model $X_{err}$ and $Z_{err}$ are uncorrelated with True, so the non-zero values in Figs. 5 – 6 are a result of the finite data set (1460 in this example). The $COV_{err}(X,Z)$ (Fig. 2) increases as $a$ increases, reaching a maximum value in the upper troposphere of about $800\%^2$ for $a=0.5$ . The terms involving True and the errors in X,Y and Z in Figs. 5 and 6 do not change in magnitude with an increasing value of $a$, but they increase in amplitude with height and are significantly large ($\sim 300\%^2$) compared to the $COV_{err}(X,Z)$. These results indicate that a large sample size



is especially important in the 2CH method; even with a sample size of 1460 the random errors caused by the neglect of the covariance terms involving True and the errors in X and Z can be significant.

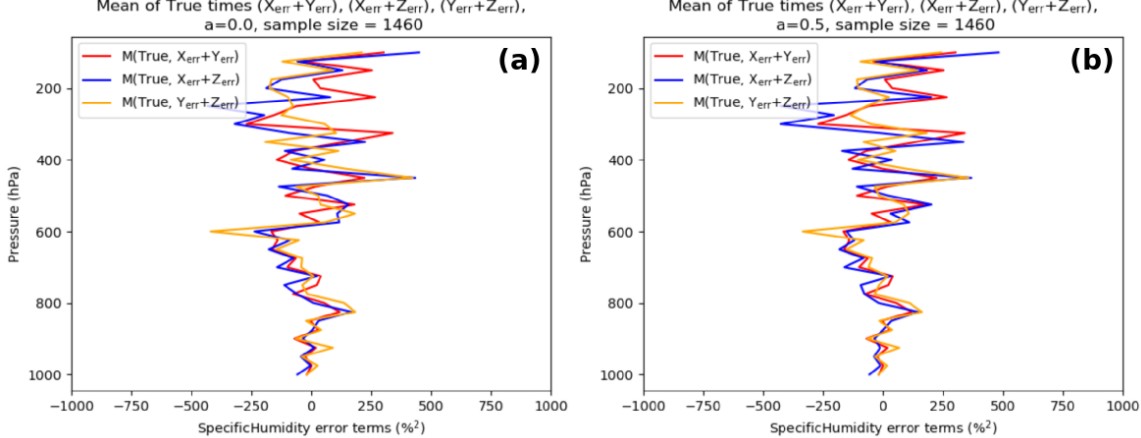

**Figure 5: Terms involving products of True and $(X_{err} + Z_{err})$, $(X_{err}+Y_{err})$, and $(Y_{err}+ Z_{err})$ for (a) $a=0$ and (b) $a=0.5$. Note that the magnitudes of True and $(X_{err}+Y_{err})$ and True and $(Y_{err}+ Z_{err})$ do not depend on the correlation between $X_{err}$ and $Z_{err}$.**

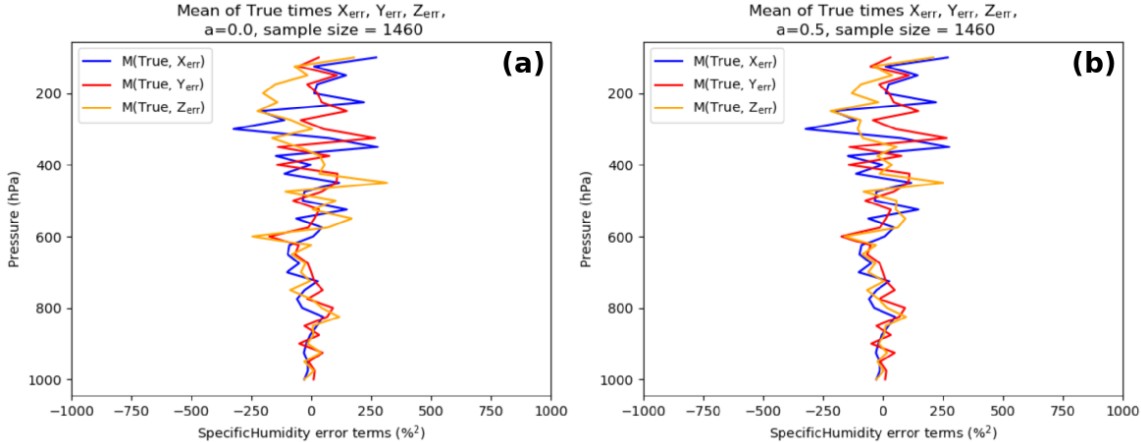

**Figure 6: Mean of products of True and $X_{err}$, $Y_{err}$, and $Z_{err}$ for (a) $a=0$ and (b) $a=0.5$.**

Figure 7 shows the exact and estimated error STD of X, Y, and Z for various combinations of the data sets and values of $a$. The estimated error STD for X, Y, and Z vary around the exact solutions (black lines) for all values of $a$ if data sets with uncorrelated errors are combined: (a) $STD_{err}(X)$ computed with Y, (b) $STD_{err}(Y)$ computed with X, and (c) $STD_{err}(Z)$ computed





with Y, and (e) STD$_{err}$(Y) using Z. Note in (c) the exact STD$_{err}$(Z) gets smaller with $a$ increasing from 0 to 1, as described previously. Figs. (d) and (f) show how correlated errors between the data sets affect the estimated error variances. Both the estimated STD$_{err}$(X) and STD$_{err}$(Z) become too small when the data sets X and Z are combined and the value of $a$ is increased. The exact solutions for all values of $a$ in (f) are in black.

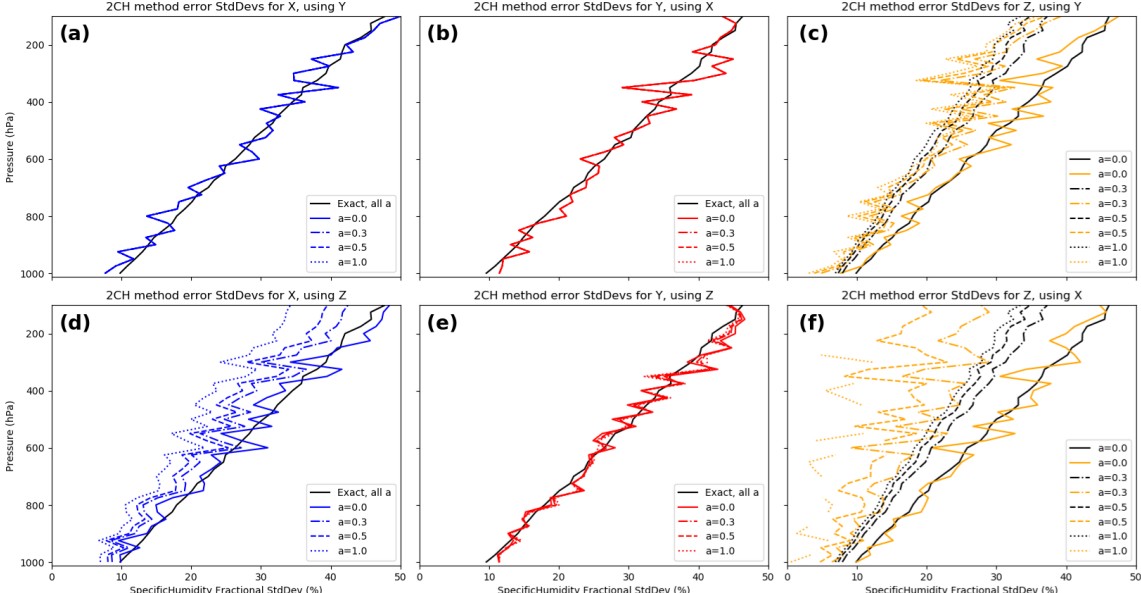

**Figure 7: Two 2CH results each for the error STD for X (a and d), Y (b and e) and Z (c and f), depending on the second data set. Profiles corresponding to several values of correlation parameter $a$ are included. The solid black profile is the exact error STD profile for X and Y for all values of $a$, and for Z when $a$=0.**

10 **5.4 Comparison of 3CH and 2CH methods using the error model**

Figure 8 compares the error estimates from the 3CH and 2CH methods for $a$=0.5. In Fig. 8a (3CH) the solid lines denote exact error STD of X, Y and Z. The exact error STD are the same for X and Y, and are smaller for Z as discussed earlier. The estimated error STD (dashed lines) are less than the exact values for X and Z and greater than the exact values for Y, also as discussed earlier. In Fig. 8b (2CH), solid lines denote a combination of data sets with uncorrelated errors (where error terms

15 are neglected, but are non-zero due to a finite sample size), while dashed lines indicate combinations of data sets with correlated errors and neglected error terms. We see that for the 3CH method and 2CH method, all estimates are similar (with the exception of STD$_{err}$(Y), which is affected in the 3CH method by correlation of X$_{err}$ and Z$_{err}$). However, the profiles of the 2CH estimates





show considerably more noise than those of the 3CH estimates, which is a consequence of the larger magnitude of the neglected error terms in the 2CH method (see Sects. 2.4 and 5 above).

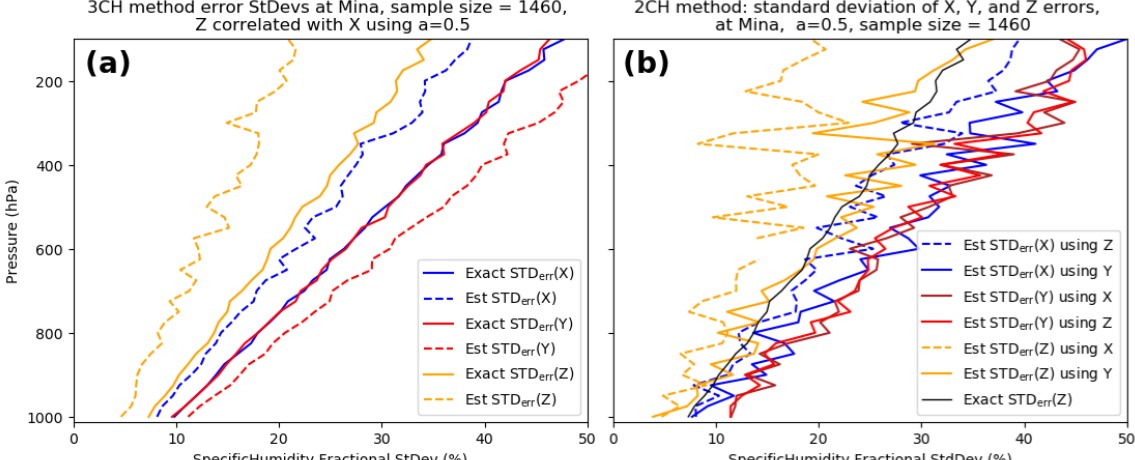

**Figure 8: Estimated and exact error standard deviations for 3CH method (a) and 2CH method (b) for a correlation of Z and X errors**
5    **of 0.45 (*a*=0.5). (a) Exact STD errors for X, Y and Z given by blue, red and orange solid lines respectively. Estimated error STD error given by dashed lines of same color. (b) Exact error STD of Z given by solid black line. Estimated error STD of X using Z and Y given by blue lines, estimated error STD of Y using X and Z given by red lines, and estimated error STD of Z using X and Y given by orange lines. For all colored lines, error terms are neglected. Solid lines indicate estimates from combinations of data sets with uncorrelated errors, dashed lines indicate estimates from combinations of data sets with correlated errors and hence larger error**
10   **terms.**

We next consider the effect of the sample size on the error estimates from the 3CH and 2CH methods. We repeat the calculations from both methods by using a subset of the 1460 samples used in the above calculations. We created the subset by selecting every tenth sample from the complete set, giving a sample size of 146.





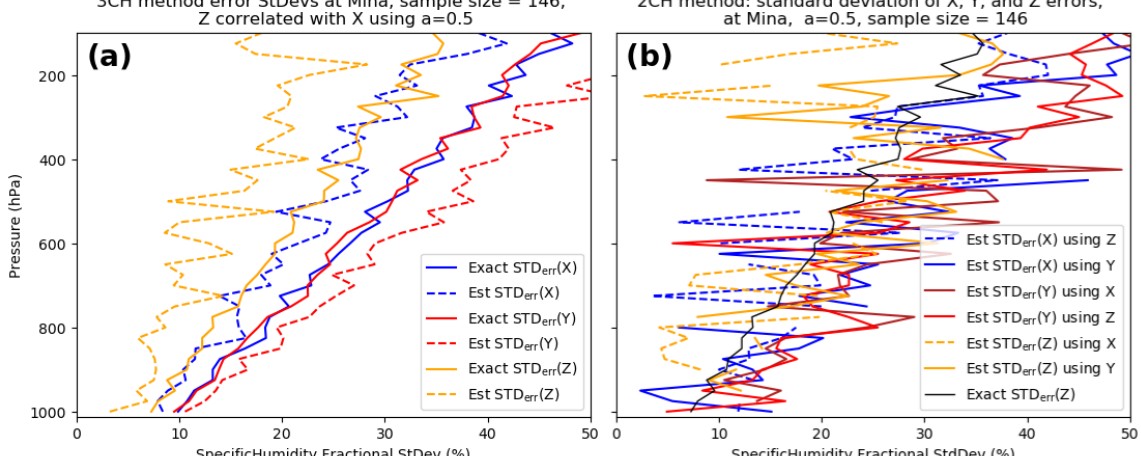

**Figure 9: Same as Fig. 8 except for a smaller sample size of 146. All profiles become much noisier, with the 2CH method showing significantly more noise than the 3CH method.**

Figure 9 shows the same estimates as those in Fig. 8, except for the much smaller sample size of 146. For the smaller sample size, the noise increases for both methods. However, the effect on the 3CH method is less than on the 2CH method, and the differences in the estimates are still clearly visible in the 3CH method. For some of our comparison data sets using real data (Anthes and Rieckh, 2018), the sample size n is less than 100 in the lower and upper troposphere; hence the noise due to small sampling size is likely to be significant in the estimates for these regions.

## 6 Estimates of error variances using 3CH and 2CH methods and real observations

Anthes and Rieckh (2018) showed a large number of error variance estimates using real data and the 3CH method at four radiosonde (RS) locations in the Pacific Ocean region. Here we show a few examples of how the 2CH method compares with these 3CH estimates.

We use co-located data of radio occultation (RO), RS, NCEP Global Forecast System (GFS), and ERA-Interim (ERA) at Minamidaitojima (Mina), which is located on Okinawa at 25.6°N 131.5°W, and is one of the four RS stations studied by Anthes and Rieckh (2018) and Rieckh et al. (2018). We use the RO-Direct method for computing RO specific humidity (uses GFS temperature to compute specific humidity $q$ from observed RO refractivity). Details about the co-location criteria and specific humidity retrieval are described by Rieckh et al. (2018). Data pairs at each level are only used if data are available for all four data sets. All data sets are interpolated to a common 25 hPa grid. In the 2CH method, Eq. (11a) is applied for various





combinations of the four data sets. Figure 10a shows the number of co-located profiles per pressure level. Figure 10b shows the normalized RS specific humidity values for these profiles.

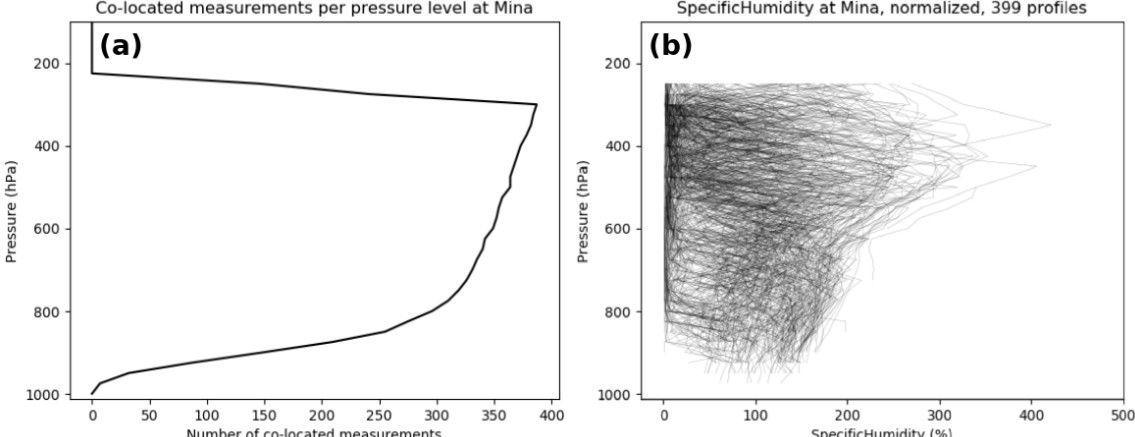

**Figure 10: (a) Number of co-located measurements (data pairs) per pressure level for RO, RS, GFS and ERA. (b) Normalized $q$**
**values for radiosondes (RS) at Mina, 2007. The normalized $q$ values are computed as $100(q - q_{ERA})$/CLIMO where CLIMO is the annual mean value of $q$ for 2007. Profiles cut off at 250 hPa because RS data are not reported at higher levels. At the bottom, co-located profiles thin out since RO penetration depth varies and only very few profiles are available at the lowest levels.**

We use our results from the 3CH method for real data (Anthes and Rieckh, 2018) to evaluate the results of the 2CH method.
Figure 11a shows the 3CH estimated error variances for specific humidity for ERA, GFS, RS and RO using three independent equations (Anthes and Rieckh, 2018). The mean value of the estimates is given by the solid line and the STD of the estimates about the mean by the shading. The 3CH estimates are considered reasonably accurate for the reasons given in Anthes and Rieckh (2018), namely that the magnitude and shape of the estimates for refractivity agree with other independent refractivity error estimates (so we assume that the method works just as well for humidity as for refractivity), and that the results are
consistent for the four different RS stations studied.

The other panels in Fig. 11 show the 2CH estimates of the error variance for ERA (b), RO (c) and RS (d) using various pairs of observations. We consider the 2CH estimates unrealistic for these data sets. The magnitudes reach values that are negative or up to five times the magnitudes of the 3CH method, which is considered unrealistically large. The profiles of the estimates
of error variances of ERA, RS and RO are also quite different depending on the pairs of observations used, unlike the 3CH method is which all combinations of observations give similar profiles. Similar results were obtained for 2CH estimates of refractivity (not shown).





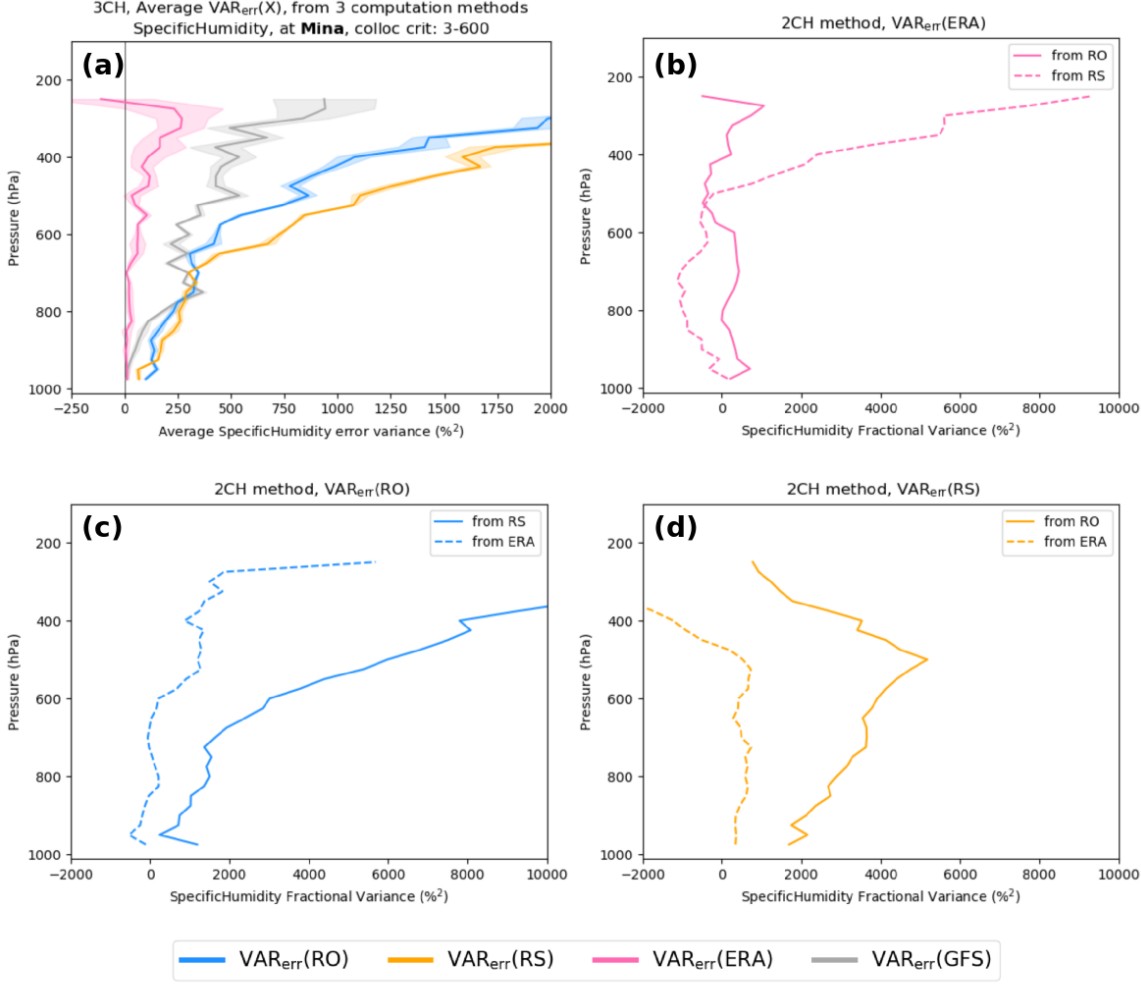

**Figure 11: Estimated error variances for specific humidity: (a) results from the 3CH method for ERA (purple), GFS (gray), RO (blue), and RS (orange). Other three panels: 2CH method estimates for ERA (b), RO (c) and RS (d).**

5   Thus we find that the 2CH method produces estimates of the error variances for specific humidity that are quite different from those of the 3CH method. This is somewhat surprising based on the comparison of the two methods using the random error model, in which the results were similar except for the greater noise in the 2CH method, a result of the larger neglected error terms in the 2CH method. We suspected that the cause for the different behavior using real data might lie in the different





treatment of bias errors in the real data in the 2CH and 3CH method. To investigate this hypothesis, we considered the effect of adding a bias error $\varepsilon$ to Z.

### 7 Comparison of 3CH and 2CH methods including a bias

#### 7.1 Effect of a bias in the error model

In order to investigate the effect of a bias in one of our data sets, we go back to the derivations of error variance terms in Sect. 2. For the 3CH method, Eq. (2a) becomes

$$VAR_{err}(X)_{est} = \tfrac{1}{2}[MS(X - Y) + MS(X - (Z+\varepsilon)) - MS(Y - (Z+\varepsilon))] \tag{19}$$

which may be expanded to

$$VAR_{err}(X)_{est} = M(X^2) - M(X^2 - XY - XZ + YZ) + \varepsilon[M(Y_{err}) - M(X_{err})] \tag{20}$$

For random errors and a large sample size the means of $X_{err}$ and $Y_{err}$ will be very small and the difference of the mean errors

will also be small. Thus the bias term will tend toward zero as the sample size increases and the effect of a bias error in Z is minimal in the 3CH method. However, this is not the case in the 2CH method, as shown below. Equation (11a) becomes

$$VAR_{err}(X)_{est} = MS(X) - [MS(X+(Z+\varepsilon)) - MS(X - (Z+\varepsilon))]/4 \tag{21}$$

which can be expanded to

$$VAR_{err}(X)_{est} = MS(X) - M(XZ) - \varepsilon M(X). \tag{22}$$

Recalling that the variables X, Y, Z and $\varepsilon$ are all normalized by CLIMO, M(X) is approximately 1 (100%) or the term $\varepsilon M(X)$

is approximately $100\varepsilon$ %².

$$VAR_{err}(X) = MS(X) - M(XZ) - 100\varepsilon(\%^2) \tag{22a}$$
where $\varepsilon$ is expressed as a percent.





For a bias error ε of 10%, the error term is 1000%$^2$, which is large compared to the true VAR$_{err}$(X). For the 2CH method a positive bias in the data set Z will cause a negative error in the computed error variance of X, and a negative bias in the data set Z will cause a positive error in the computed error variance of X.

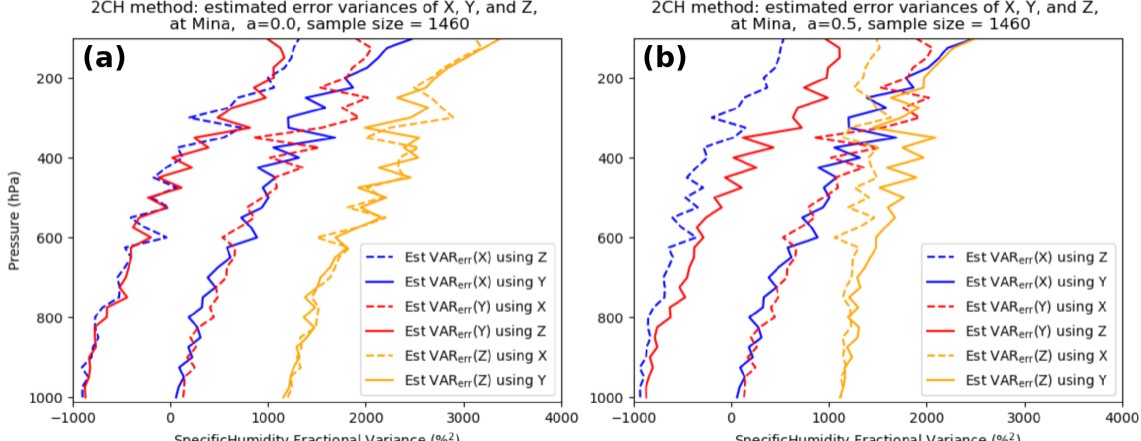

**Figure 12: Effect of adding a constant bias of 10% to Z for no correlation between X and Z errors (a) and *a*=0.5 (b). For *a*=0, the estimated error variance of Z is increased (orange), and the estimated error variances for X and Y computed with Z (blue dashed and red solid) are erroneously low. Adding correlation of X and Z errors to the bias in (b) creates an even more complicated picture.**

Figure 12 shows the effect of adding a constant bias of 10% to Z for no correlation of random errors (*a*=0) and positive correlation between random errors of X and Z given by *a*=0.5. The correct error variance profiles are given by the solid blue (VAR$_{err}$(X) using Y) and dashed red profiles (VAR$_{err}$(Y) using X). For no correlation of random errors (Fig. 12a), the effect of adding a constant bias of 10% to Z is to produce estimated error variances of X using Z and Y using Z that are much too low. Conversely, the estimates of error variance of Z using X or Y are much too high. When the random errors of X and Z are correlated (Fig. 12b), similar bias errors in the estimated profiles involving Z are evident, but the correlation produces more noise in the upper troposphere.

### 7.2 Simulating the observed real data bias in the 2CH method

To see if a bias in our real data could explain the very different estimates of the error variances shown in Fig. 11b – 11d for the 2CH method, we set up empirically based bias profiles in the simulated data. These match approximately the observed differences of RS and RO from ERA as found by Rieckh at al. (2018) in the real data sets (supplement, Figure S5 panel 4 for RS; Figure S6 panel 1 for RO). For the tests we consider ERA the truth. We computed the specific humidity annual mean





biases of RO and RS for each level as 100[Mean(RO) − CLIMO]/CLIMO and 100[Mean(RS) − CLIMO]/CLIMO, respectively. We used these results to create a simple mean bias for both Y and Z. The biases are depicted in Fig. 13 (dashed lines), along with the real RS and RO annual mean biases (solid lines). More specifically, the bias used for Y (simulating the RO bias) varies linearly between pressure levels as:

5      −5 to 2 % from 1000 to 800 hPa

2 to −4 % from 800 to 650 hPa

−4 to 2 % from 650 to 550 hPa

2 to 0 % from 550 to 250 hPa.

10     The bias used for Z (simulating the RS bias) varies linearly between pressure levels as:

to 5 % from 1000 to 800 hPa

to −5 % from 800 to 500 hPa

−5 to −55 % from 500 to 250 hPa.

The respective bias is added to both Y and Z when the data are created from the True profiles:

$X = \text{True} + X_{err}$; corresponds to ERA

$Y = \text{True} + Y_{err} + Y_{bias}$; corresponds to RO

$Z = \text{True} + Z_{err} + Y_{bias}$; corresponds to RS

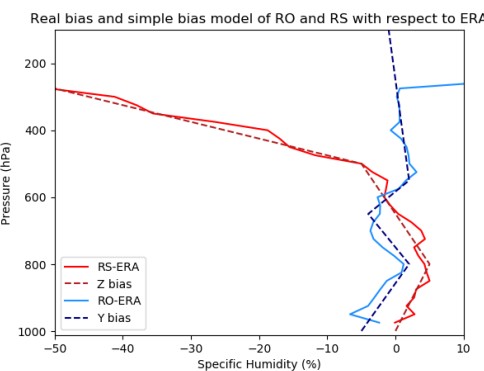

**Figure 13:  Annual mean normalized profiles of RS − ERA (solid red) and RO − ERA (solid blue) and empirical bias profiles (dashed lines) based on the mean profiles.**

We use these specified biased data sets to compute error variances via the 2CH method. Results are shown in Fig. 14, with the

results of the real data in the top row (a − c), and the results of the simulated bias data in the bottom row (d − f). The error



variances for X (Fig. 14d) look similar to its corresponding real data set ERA (Fig. 14a). The error variance estimates for X using Y (solid) versus Z (dashed) agree in their overall shape to ERA using RO (solid) versus RS (dashed). Since our empirical bias model is very simple, agreement between X and ERA estimates is not perfect. Similarities between the real (top) and simulated (bottom) estimates can also be seen for RO and Y (Figs. 14b and 14e). Differences between the real and simulated

data results are largest for RS and Z (Figs. 14c and 14f), especially when RS and RO (Z and Y) are combined (solid lines).

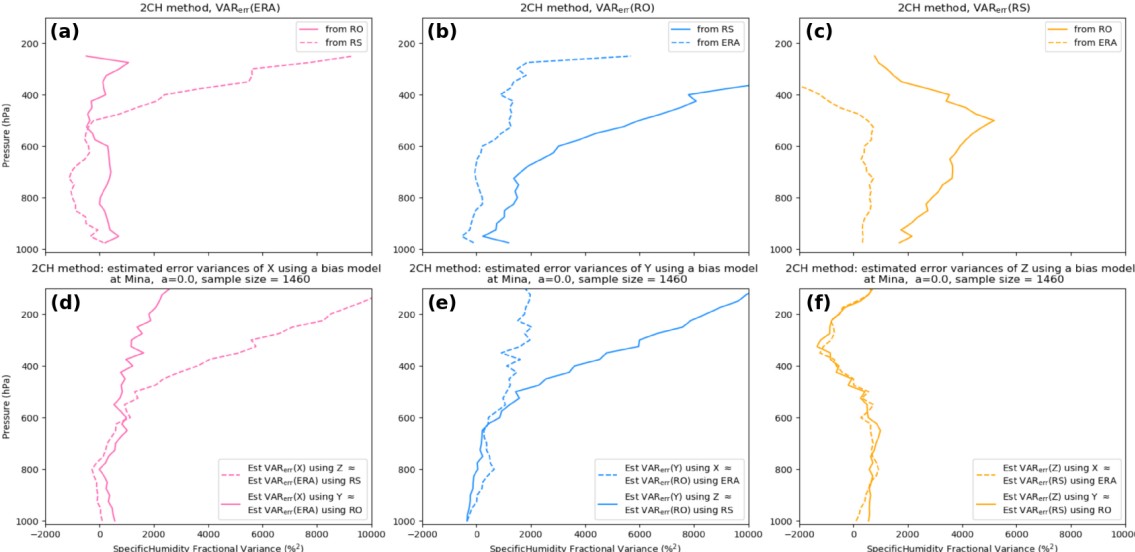

**Figure 14: 2CH method error variances for ERA, RO, and RS (top row) and for simulated data using the specified empirical bias profiles (bottom). Correlation between X and Z errors is zero for this experiment.**

## 8 Summary and Conclusions

In this study we compared two methods for estimating the error variances of multiple data sets, the three-cornered hat (3CH) and two-cornered hat (2CH) methods. Using a specified error model in which we could vary the degree of correlation between two data sets as well as specifying bias errors, we examined the sensitivity of the 3CH and 2CH methods to random and bias errors. For the error model, we added known random or bias errors to 1460 specific humidity profiles (considered truth) obtained from the ERA-Interim reanalysis over a subtropical radiosonde station in Japan. We compared the effect of neglecting

various error covariance and other error terms in the 3CH and 2CH methods on the estimated error variances and standard deviations. We also considered the effect of a finite sample size on the estimates by repeating the calculations using a subset of 146 of the total 1460 profiles. We found that the 3CH method was less sensitive to the neglected error terms for various random error correlations than the 2CH method.




We also compared the 3CH and 2CH methods using real radiosonde (RS) and radio occultation (RO) data, as well as GFS model data. We find that the 3CH method produces more consistent and accurate results than the 2CH method when using real data. The 2CH method produced very different estimates of the error variance of ERA depending on which observational data

set (e.g. RS or RO) was used in the comparison. Using an empirical bias model based on observed RS and RO difference from ERA during 2007, we showed that these differences in error variance estimates were likely caused by different biases in the RS and RO data. The effect of bias errors is shown to give unrealistic results using the 2CH method.

*Code availability.* Code will be made available by the authors upon request.

*Data availability.* Data can be made available from the authors upon request.

*Author contributions.* Both authors contributed equally to the ideas and conceptual development. The first author computed the results.

*Competing interests.* The authors declare that they have no conflict of interest.

*Acknowledgments.* The authors were supported by NSF-NASA grant AGS-1522830. We thank Eric DeWeaver (NSF) and Jack Kaye (NASA) for their long-term support of COSMIC.

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
