# Peer review of "Evaluating two methods of estimating error variances using simulated data sets with known errors"

_Atmospheric Measurement Techniques, 2018_

## Referee Comment (RC1) · Anonymous Referee #1 · 26 Mar 2018

The paper compares two simple methods for estimating error variances from double or triple collocated data using simulated and real data. The error estimation methods are the two and three cornered hat methods (2CH and 3CH). The paper is interesting, but should be considerably improved. The present version is not suitable for publication.

The presentation is rather sloppy, and it looks more like a report than a scientific paper. For instance, equations (18b-c-d) are essentially the same equation, and indicating cancelling terms by crossing them out (pages 11 and 12) is not the style of a scientific paper. Variance is defined, but MS (equation (2) and further) not.

More seriously, the authors seem to assume that their data are well calibrated and do

not contain any representativeness errors - at least these problems are nowhere mentioned. Therefore they are surprised when pairwise 2CH error estimates are different from each other and from 3CH error estimates, attribute this to biases, and work this idea out in section 7. But the differences may very well be a matter of representativeness. Both in 2CH and 3CH the "true" signal is the common signal shared by the two or three observation systems under consideration, and the "true" signal is determined by the system with the lowest resolution, see Stoffelen (1998). The authors should consider this.

Another serious point is that the authors assume their error model to be valid without any further justification. A scatter plot of the data - the starting point of all analysis of data from multiple sources - would be helpful here. It will show if any calibration issues play a role and if errors indeed can be assumed to be independent of the observed value. The current presentation is more of a "trial and error" type. This should be considerably improved.

A minor point: the link to Wriley (2003) gives no information on the history of the 3CH method.

---

## Referee Comment (RC2) · Anonymous Referee #2 · 27 Mar 2018

**General comments**

The authors compare error variances as generated from the three-cornered hat (3CH) and two-cornered hat (2CH) methods from simulated and real data sets. As expected, the 3CH results were less noisy and less sensitive to biases. The authors do a good job of presenting their widespread findings (from simulated and correlated data, as well as collocated real observations); however, I have some comments and suggestions that I would like to see addressed in the their revision.

The 3CH assumes independent, uncorrelated observations, and its usefulness can be

limited by the sample size, as well as the variability the source data itself. The authors do mention several times that finite sample sizes will cause the cross-correlation terms to be non-zero. However, they do not seem to address the point that the error variances produced by the 3CH method (and likely 2CH as well) can be dominated by a data source that is largely different than the other two in the trio being analyzed. Thus, the size of the relative errors from the RO, GFS, ERA, and RS observations comes into play in the accuracy of the 3CH results in Section 6.

Some of the equation development is either incomplete or hard to follow. For example, on Pages 11-12, the authors derive equations for the estimated error variance of X,Y,Z. They start with the traditional equation for the variance (equations 2,3, and 4), but cross out some terms, then neglect the covariance term in the next step (because it's the 3CH estimate), then plug it back in. It took me a while to put it all together, so maybe the authors can add some additional text or format differently to help the reader along.

The authors also seem to already possess knowledge regarding climatological processes and models to set up and normalize their model profile (section 3.1). Perhaps these numbers are taken from some of their previous work, but some additional text or references pertaining to the source or reasoning behind these numbers would be helpful.

**Line edits**
The grammar and sentence structure is quite good and easy to read, I only have a few line edits to offer.
Line 11: Should be W.J. Riley, not W.J. Wriley
Line 13: 3CH, not 3HC

Line 17: Should x,z be capitalized?
Line 3: Should be There, not These

---

## Referee Comment (RC3) · J. Vogelzang (Referee) · 28 Mar 2018

**Some additional remarks**

I tried the 2CH method on a triple collocated data set that I have available. The dataset consists of zonal and meridional ocean surface wind components (*u* and *v*), measured by buoys (not blacklisted by ECMWF), ASCAT-A coastal, and predicted by ECMWF. The data cover January 2015.

The results for the error standard deviations are listed in the table below. The triple collocation calculation (TC) was done using the scatterometer as calibration reference and without taking representativeness errors into account.

| Model | scatterometer | | buoys | | ECMWF | |
|---|---|---|---|---|---|---|
| | $\sigma_u$ | $\sigma_v$ | $\sigma_u$ | $\sigma_v$ | $\sigma_u$ | $\sigma_v$ |
| TC | 0.56 | 0.79 | 1.20 | 1.15 | 1.45 | 1.44 |
| 2CH | 0.43 | 0.76 | 1.27 | 1.19 | -- | -- |
| 2CH | 1.31 | 1.60 | -- | -- | 0.86 | -0.40 |

The table shows that the 2CH results for scatterometer and buoy agree well with the TC results, but the 2CH scatterometer and ECMWF results do not. This can be explained by calibration issues. The ASCAT scatterometer has been calibrated carefully with respect to buoys, and the calibration scalings are 1.000 for the zonal wind and 1.004 for the meridional wind. The calibration scalings for the ECMWF model w.r.t. the scatterometer are 0.967 and 0.946. These numbers appear close to 1, but their effect on the error estimates is considerable.

In the 2CH method, the error variance of system 2 is essentially the difference between system's 2 autocovariance minus its cross-covariance with system 1. For the scatterometer example above, covariances and cross variances are between 30 and 45. Lets assume it is 40 for the autocovariance and 39 for the cross-covariance, the difference being an error variance of 1 $m^2s^{-2}$. If we now scale the system 2 data by $a = 1.01$, a difference of 1% in the calibration scaling, the autocovariance scales with $a^2$ and becomes 40.804, while the cross covariance scales with $a$ and becomes 39.39. The difference, the error variance in the 2CH model, now becomes 1.414. So a 1% difference in calibration scaling leads to a difference of more than 40% in the error variance estimate in this example.

In the 2CH method the error variances are the difference between two large numbers. Therefore the method is very sensitive to calibration and representativeness issues.

Jur Vogelzang, KNMI

---

## Short Comment (SC1) · 7 May 2018

We thank Dr. Vogelzang for his interesting comment and illustrative example using the 2CH method, which we interpret as supporting our conclusion in the paper that the 2CH method is very sensitive to bias errors. As he says, the agreement of the 2CH method using the scatterometer and buoy data with the results from the TC method is likely because the scatterometer and the buoy data have only small biases with respect to each other (calibration scalings very close to 1.0). However, the use of ECMWF data with the scatterometer data produces large errors using the 2CH method because the ECMWF data have larger biases with respect to the scatterometer data (calibration

scaling ∼0.95). Thus these biases cause the large errors in the 2CH method when using scatterometer and ECMWF data sets (as shown in his Table) and calibration of the two data sets is necessary to eliminate the biases when using the 2CH method.

---

## Author Response (AR1)

We thank the two referees for their comments and constructive feedback, which we have considered in the revised paper. Their comments (in italics) and our responses are addressed below. All page and line numbers in the referee's comments refer to the originally submitted version of the paper. All page and line numbers in our response refer to the revised paper.

**Anonymous Referee #1**
*The paper compares two simple methods for estimating error variances from double or triple collocated data using simulated and real data. The error estimation methods are the two and three cornered hat methods (2CH and 3CH). The paper is interesting, but should be considerably improved. The present version is not suitable for publication.*

We have revised the paper according to the referee's suggestions.

1. *The presentation is rather sloppy, and it looks more like a report than a scientific paper. For instance, equations (18b-c-d) are essentially the same equation, and indicating cancelling terms by crossing them out (pages 11 and 12) is not the style of a scientific paper. Variance is defined, but MS (equation (2) and further) not.*

   We have fundamentally revised the structure of the paper, eliminated some of the equations and intermediate steps of the derivations, moved the 2CH derivation into an appendix, addressed the style issue, and made sure all the terms are defined. We have also eliminated the need for both lower case and upper case X, Y and Z.

2. *More seriously, the authors seem to assume that their data are well calibrated and do not contain any representativeness errors - at least these problems are nowhere mentioned. Therefore they are surprised when pairwise 2CH error estimates are different from each other and from 3CH error estimates, attribute this to biases, and work this idea out in section 7. But the differences may very well be a matter of representativeness. Both in 2CH and 3CH the "true" signal is the common signal shared by the two or three observation systems under consideration, and the "true" signal is determined by the system with the lowest resolution, see Stoffelen (1998). The authors should consider this.*

   The reviewer raises issues related to *calibration* and *representativeness errors* in this comment.

   *Calibration:* The calibration issue is closely related to the bias issue that we discuss. In the TC method, two of the data sets are calibrated against a third so that the three data sets are not biased with respect to each other. In the simplest form of the 3CH method, the data sets are not calibrated against each other, and thus some biases can exist. We investigate the effect of a 10% bias in one of the data sets and show even this fairly large bias does not greatly affect the 3CH results. However, even a much smaller bias affects the 2CH in a significant way. In addition, we also investigated the effect of biases in the related paper Anthes and Rieckh, 2018 by comparing the 3CH method to the triple co-location (TC) method (Stoffelen, 1998; Vogelzang et al., 2011). The results using the TC, in which the data sets were calibrated with respect to each other, were

very similar to those using the 3CH method, confirming our results with the error model that biases do not cause large errors in the 3CH method.

To address this comment, we have added the following discussion to the beginning of Section 5:

"All four data sets have some degree of unknown bias for certain locations, altitudes, or atmospheric conditions; none of them represent the ultimate "truth" and there is no standard atmospheric data set for calibration. However, they have all been compared to other models or observations to one degree or another. We investigated the effect of biases in the related paper Anthes and Rieckh (2018) by comparing the 3CH method to the triple co-location (TC) method (Stoffelen, 1998; Vogelzang et al., 2011). The results using the TC method, in which the data sets were calibrated using the ERA-Interim data set as the calibration reference, were very similar to those using the 3CH method."

2) *Representativeness errors*: We mention representativeness errors in the Introduction and discuss them in Section 5. The idealized model data sets with prescribed errors we developed are free of representativeness errors; they contain only known (specified) random and bias errors. However, a portion of the specified errors in the error model may be thought of as representativeness errors. Also, the 3CH method includes representativeness errors as part of the error estimates.

The two model (ERA-Interim and GFS) and two observational (RS and RO) data sets include random and bias errors. Representativeness errors, which are included in the error estimates from both the 2CH and 3CH method, may contribute to these random or bias errors. As our results show, random errors do not cause large differences between the 2CH and 3CH methods if the sample size is large enough; only bias errors do.

We showed that biases in one of the two data sets with respect to the other will cause large errors in the 2CH method. We believe this is the issue that Dr. Vogelzang discussed in his supplement discussion. We agree with his conclusion, that the 2CH method is very sensitive to calibration (bias) issues.

We added the following paragraph on page 16 to address the comment on representativeness errors:

 "The two model and the RO data sets are representative of similar horizontal scales (~100 km), while the radiosonde data are in-situ point measurements and therefore represent a much smaller horizontal scale. However, many studies (e.g. Ho et al., 2010a,b; Kuo et al. 2004, Chen et al. 2011) have used radiosonde data as correlative data for verifying models, RO, and other data sets without applying corrections for representativeness errors. These results indicate that the different representative scales are not a significant source of error in the comparisons (unlike spatial and temporal sampling errors resulting from the time and spatial differences between the data sets, which we correct for). However, any representativeness errors are included in the error estimates using either the 2CH or 3CH method."

3. *Another serious point is that the authors assume their error model to be valid without any further justification. A scatter plot of the data - the starting point of all analysis of data from multiple sources - would be helpful here. It will show if any calibration issues play a role and if errors indeed can be assumed to be independent of the observed value. The current presentation is more of a "trial and error" type. This should be considerably improved.*

The error model is based on previous studies that have estimated the error profile of specific humidity. In order to justify the error model, we have added the following paragraph to the end of Section 3, page 5 line 19:

"The error model is created based on error estimates of specific humidity from several studies (e.g. Kursinski, 1997; Collard and Healy, 2003; von Engeln and Nedoluha, 2005; Wang et al., 2013). For example, Collard and Healy (2003) found that, for tropical conditions, the percentage errors for RO specific humidity varied from approximately 10% near the surface to about 70% near 300 hPa. Other studies show the errors varying from about 10% near the surface to 100% in the upper troposphere (about 200 hPa). In our error model, we specified the STD to roughly approximate the STD of RO or RS data at Minamidaitojima, Japan (Anthes and Rieckh, 2018; Rieckh et al., 2018). The assumed STD of normalized q (percent error) given by Eq. (9) is consistent with the above empirical error estimates. Thus the error model is a reasonable one in terms of its magnitude and increase with height.  Since it is intended to show the sensitivity of the 3CH and 2CH methods to varying degrees of correlation between two of the data sets used in the comparison, it is not necessary that the error model be a close replication of any particular observing system, just that the magnitude of the assumed errors and their vertical distribution be reasonable.

4. *A minor point: the link to Wriley (2003) gives no information on the history of the 3CH method.*

Thank you; we have revised this sentence to:
"W.J. Riley (2003), and references therein, provides a summary of the 3CH method."

**Anonymous Referee #2**
*General comments*

*The authors compare error variances as generated from the three-cornered hat (3CH) and two-cornered hat (2CH) methods from simulated and real data sets. As expected, the 3CH results were less noisy and less sensitive to biases. The authors do a good job of presenting their widespread findings (from simulated and correlated data, as well as collocated real observations); however, I have some comments and suggestions that I would like to see addressed in their revision.*

*The 3CH assumes independent, uncorrelated observations, and its usefulness can be limited by*

*the sample size, as well as the variability the source data itself. The authors do mention several times that finite sample sizes will cause the cross-correlation terms to be non-zero. However, they do not seem to address the point that the error variances produced by the 3CH method (and likely 2CH as well) can be dominated by a data source that is largely different than the other two in the trio being analyzed. Thus, the size of the relative errors from the RO, GFS, ERA, and RS observations comes into play in the accuracy of the 3CH results in Section 6.*

Page 2 line 23 says that widely different errors associated with the three systems can reduce the accuracy of the estimates. We have inserted references to this in the revised paper. In our error model, the magnitudes of the errors in all three simulated data sets are similar. Previous studies of the errors of the systems tested in our paper (two model sets, ERA-Interim and GFS, and two observational data sets, RO and RS), indicate that the errors of all these systems are also of similar orders of magnitude (please see references in the paper and below). In addition, we recently used the 3CH to estimate the errors associated with these data sets and found that they were similar in magnitude (Anthes and Rieckh, 2018).

*Some of the equation development is either incomplete or hard to follow. For example, on Pages 11-12, the authors derive equations for the estimated error variance of X,Y,Z. They start with the traditional equation for the variance (equations 2,3, and 4), but cross out some terms, then neglect the covariance term in the next step (because it's the 3CH estimate), then plug it back in. It took me a while to put it all together, so maybe the authors can add some additional text or format differently to help the reader along.*

We agree and have revised the presentation of the equations by reducing the number of similar equations and clarifying the presentation. Please see also the response to Comment 1 of Reviewer 1.

*The authors also seem to already possess knowledge regarding climatological processes and models to set up and normalize their model profile (section 3.1). Perhaps these numbers are taken from some of their previous work, but some additional text or references pertaining to the source or reasoning behind these numbers would be helpful.*

We have provided considerably more discussion in the first part of Section 3 on the error model and its properties, including comparison with other studies of the estimated errors of specific humidity observations in the atmosphere. Please see our response to point 3 of Reviewer #1 (above) and the added paragraph discussing the error model.

*Line edits*
*The grammar and sentence structure is quite good and easy to read, I only have a few line edits to offer.*

Thank you for the careful reading. We have corrected all these typos.

*Page 2*
*Line 11: Should be W.J. Riley, not W.J. Wriley*

Corrected.

*Line 13: 3CH, not 3HC*
Corrected.

*Page 8*
*Line 17: Should x,z be capitalized?*

Section 3 had confusing notation with the upper and lower case variables. We have revised the text and equations to use only upper case X, Y and Z.

*Page 9*
*Line 3: Should be There, not These*
Corrected.

References (also included in revised paper)

Anthes, R.A. and Rieckh, T: Estimating observation and model error variances using multiple data sets. Atmos. Meas. Tech. Discuss., https://doi.org/10.5194/amt-2017-487, 2018.

Chen, S.-Y., Huang, C.-Y., Kuo, Y.-H., and Sokolovskiy, S.: Observational Error Estimation of FORMOSAT-3/COSMIC GPS Radio Occultation Data, Mon. Wea. Rev., 139, 853–865, https://doi.org/10.1175/2010MWR3260.1, 2011.

Collard, A.D. and Healy, S.B.: The combined impact of future space-based atmospheric sounding instruments on numerical weather-prediction analysis fields: A simulation study, Q. J. R. Meteorol. Soc., 129, pp. 2741–2760 doi: 10.1256/qj.02.124, 2003.

[revised manuscript text omitted]

$$VAR_{err}(Y)_{est} = {}^1\!/_2\left[MS(X-Y) + MS(Y-Z) - MS(X-Z)\right] \tag{4a}$$

$$VAR_{err}(Z)_{est} = {}^1\!/_2\left[MS(X-Z) + MS(Y-Z) - MS(X-Y)\right] \tag{5a}$$

5  ## 2.2  Two-cornered hat (2CH) method

 The 2CH method  uses only two data sets, X and Z.  The derivation is presented in Appendix A. The 2CH method error variances for X and Z

10  $$\sum(X+Z)^2 = 4\sum True^2 + \sum\left[(X_{err}+Z_{err})^2 + 4True(X_{err}+Z_{err})\right]$$

$$MS(X+Z) = 4MS(True) + VAR_{err}(X) + VAR_{err}(Z)$$
$$+ 2COV_{err}(X,Z) + 4M(True, X_{err}+Z_{err})$$

15

20

 are given as

$$VAR_{err}(X) = MS(X) - {}^1\!/_4\left[MS(X+Z) - MS(X-Z)\right]$$
$$- 2M(True, X_{err}) + COV_{err}(X,Z) + M(True, X_{err}+Z_{err}) \tag{6}$$

and

$$VAR_{err}(\text{}Z) = \text{}MS(Z) - {}^1\!/_4\left[MS(X+Z) - MS(X-Z)\right]$$

[revised manuscript text omitted]

$$VAR_{err}(Y) = {}^1\!/_2[MS(X - Y) + MS(Y - Z) - MS(X - Z)]$$
$$\underline{+ COV_{err}(X, Y) + COV_{err}(Y, Z) - COV_{err}(X, Z)}$$

5

$$VAR_{err}(Y)_{est} = MS(X - Y) + MS(Y - Z) - MS(X - Z) \tag{3a}$$

(4a) can be expressed as

$$VAR_{err}(Y)_{est} = VAR_{err}(Y) + COV_{err}(X, Z). \tag{3b}$$

Substituting for the COV term from Eq. (16) and noting that, for our error model, $VAR_{err}(X) = VAR_{err}(Y)$ we obtain

10 $$VAR_{err}(Y)_{est} = [(1 + 2a)/(1 + a)]VAR_{err}(Y) \tag{3c}$$

Thus the estimated error variance for Y is always greater that the true value for $a > 0$, which is seen in Fig. 3b.

Lastly, we consider the effect of the X and Z correlation on the estimate for the Z error variance. Eq.

$$VAR_{err}(Z) = {}^1\!/_2[MS(X - Z) + MS(Y - Z) - MS(X - Y)]$$
$$+ COV_{err}(X, Z) $$

$$\text{VAR}_{\text{err}}(Z)_{\text{est}} = \text{MS}(X - Z) + \text{MS}(Y - Z) - \text{MS}(X - Y) \tag{4a}$$

(5a) can be expressed as

$$\text{VAR}_{\text{err}}(Z)_{\text{est}} = \text{VAR}_{\text{err}}(Z) - \text{COV}_{\text{err}}(X, Z). \tag{4b}$$

5    Substituting for the COV term from Eq.  (16) and using Eq.  (15) we obtain

$$\text{VAR}_{\text{err}}(Z)_{\text{est}} = [(1 - a)/(1 + a^2)]\text{VAR}_{\text{err}}(Z) \tag{4c}$$

so that the estimated error variance for Z is less than the true value for $a > 0$, which is illustrated in Fig. 3c. Finally, for $a > 1$, Eq.  (4c) shows that the estimated error variance of Z is negative and the STD is undefined. For $a = 1.0$, the estimated error variance of Z is zero for an infinite data set, but oscillates around zero because of our finite data set. Thus the estimated STD

10    of $Z_{\text{err}}$ is undefined at some levels (Fig. 3c).

~~(a) Estimated standard deviation of $X_{\text{err}}$ for values of $a = 0, 0.3, 0.5, 1.0$ and 100 computed from Eq. ??. Exact STD computed from data set X (solid black profile). (b) Same as (a) except for estimated STD of $Y_{\text{err}}$. (c) Same as (a) except for estimated STD of $Z_{\text{err}}$. Note that for different values of $a$, the exact STD of $X_{\text{err}}$ and $Y_{\text{err}}$ are always the same. The exact STD of decreases $Z_{\text{err}}$ as $a$ increases for $0 < a < 1.0$. For $a > 1.0$, the exact STD of $Z_{\text{err}}$ increases as $a$ increases, becoming equal~~

15

**4.2    Summary of error correlations on 3CH method**

20

 For the 3CH method, correlation between the data sets X and Z has the following affect on their computed error variances (when covariance terms are neglected):

$\text{VAR}_{\text{err}}(X$ $)_{\text{est}} < \text{VAR}_{\text{err}}(X$$_{\text{true}}$

25    $\text{VAR}_{\text{err}}(X$

 $Y)_{\text{est}} > \text{VAR}_{\text{err}}(Y$$_{\text{true}}$

$\text{VAR}_{\text{err}}(X$ $Z)_{\text{est}} < \text{VAR}_{\text{err}}(Z)_{\text{true}}$

[revised manuscript text omitted]

---

## Referee Report (RR1)

**Second review of "Evaluating two methods of estimating error variances from multiple data sets using an error model" by Rieckh and Anthes.**

The authors have addressed most of my comments to my satisfaction, but I still have some points left.

(1) In their response to calibration issues, the authors neglect scaling errors, i.e., the term $a$ in $x_{cal} = ax + b$, with $b$ the calibration bias. This may also be important, as was raised by Vogelzang in his supplement discussion. I recommend the authors to include this.

(2) I recommend to omit paragraph 3.3 and move relevant information to the rest of the paragraph. If the authors wish, they can start paragraph 3 with a short outline.

(3) Page 1, line 3: I suggest "Both methods assume that the data sets are well intercalibrated and that the errors are uncorrelated"

(4) Page 1, line 5: the reference to Braun et al. (2001): I don't know the exact style requirements of AMT, but is a reference in the abstract allowed? I leave that to the editor.

(5) Page 1, line 17: "is" instead of "are" (subject is "estimating")

(6) Page 1, line 18: "is" instead of "are".

(7) Page 2, line 17: I suggest "estimate both errors and linear calibration coefficients of surface winds"

(8) Line 32: "X,Y, and Z"

(9) Page 5, line 4 and Page 6, line 2: Both start with "We first generate". I recommend to change line 2 of page 6 in something like "Next, we generate"

(10) Page 19, line 3: I recommend "simulated data" instead of "a specified error model".

---

## Referee Report (RR2)

Journal: AMT
Title: Evaluating two methods of estimating error variance from multiple data sets using an error model
Author(s): Therese Rieckh and Richard Anthes
MS No.: amt-2018-75
MS Type: Research article
Iteration: Revised Submission

**General Comments**

The authors did a good job of addressing my concerns with the previous version of this paper. The paper itself reads more as a well-developed, technical paper. They also added in some supplementary discussion and references to help the reader understand the development and justification of their error model.  I recommend for publication, subject to the following very minor line edits.

**Line Edits**

Again, the paper structure is good and reads very well.  I only noticed a few errors and offer the following corrections.

- Line 15: Should it read $VAR_{err}(z) <= VAR_{err}(X)$?

- Caption for Figure 3 (c): … The exact STD of "$Z_{err}$ decreases" as $a$ increases…

---

## Author Response (AR2)

**Response to Associate editor comments on AMT 2018-75 "Evaluating two methods of estimating error variances using simulated data sets with known errors"**

Note that the title has been revised to make it clear that we are using a simulated data set with known errors. This is also in response to the second Reviewer's comment number (10).

**Associate Editor comments in italics are given below, followed by our responses:**

*The manuscript compares three geophysical measurement systems in a triple collocation method, defined as 3CH, and in a pair-wise comparison for humidity variables. It remains very useful to stress the limitations and potential interpretation errors caused by pair-wise comparisons. The editor however also notes a few minor issues for consideration.*

*First, Stoffelen (1998) noted that three geophysical systems generally do not represent the same spatio-temporal average and not exactly the same variable. In order words, the true geophysical state may be sampled in more detail by two of the systems with respect to the third. This leads to a correlation of the representativeness error of the two systems, which he quantifies to affect the results for his wind analysis, later repeated by other authors for other geophysical comparisons. O'Caroll et al. (2008) recognize this potential problem, but state on representativeness for their SST analysis: "In this paper, we proceed tentatively on the assumption that these covariances are negligible, but we also make analyses to determine if this assumption is valid.". Hence, the TC equations of O'Caroll et al. simplify to what is now in this manuscript defined as 3CH. The current manuscript provides useful tests on another geophysical set and provides some analysis. However, on page 2, line 22, it is currently stated that "The major assumption in all of the above methods is that the errors of the three systems are uncorrelated" and "Correlation between any or all of the three measurement systems will reduce the accuracy of the error estimates". In fact, the in this statement implied TC method proposed by Stoffelen (1998) does take account of the correlation in the spatial representativeness error and corrects for its covariance. Hence, he does not assume uncorrelated error in all three systems as stated here. It further follows that in TC only UNKNOWN error correlation will reduce the accuracy of the error estimates, as in fact demonstrated in this manuscript too. Please revise these sentences to reflect these comments.*

We have revised the discussion on page 2 to address this comment. We moved the paragraph starting with "The major assumption of …" to immediately follow the paragraph on the 3CH method above to make it clear that "all" refers to the 3CH method. We have added the following sentences to the paragraph describing the triple collocation (TC) method: "Stoffelen (1998) discussed the correlation between part of the representativeness errors of two of the data sets and subtracted the variance common to the scale of these errors from the estimated error variance of one of the data sets used in his pairwise estimate of error variances. This correction requires an independent estimate of the correlated part of the representativeness error. All variations of TC methods assume that, apart from the correlated part of the representativeness errors, the errors of the different observation systems are uncorrelated."

*Page 3 line 18: "The last three covariance terms in Eqs. (3)–(5) are unknown for real data sets". Given the above, the statement is not generally valid. Stoffelen (1998) and others found methods to estimate these covariances for real data sets. A revision of the sentence to make it less general seems appropriate.*

Response: We have clarified this by writing "The last three covariance terms in Eqs. (3)-(5) are unknown for the real data sets, unless they are estimated independently, and are neglected when estimating the error variances..".

*In addition, note that Figures 3 and 5-14 are illegible and will need larger fonts.*

Response: we have revised the figures by increasing the size of the fonts.

**First Reviewer comments (in italics) and responses:**

*General Comments*
*The authors did a good job of addressing my concerns with the previous version of this paper. The paper itself reads more as a well-developed, technical paper. They also added in some supplementary discussion and references to help the reader understand the development and justification of their error model. I recommend for publication, subject to the following very minor line edits.*

Response: Thank you for these positive comments.

*Line Edits*
*Again, the paper structure is good and reads very well. I only noticed a few errors and offer the following corrections.*
*Page 7 Line 15: Should it read VARerr(z) <= VARerr(X)?*

Response: Yes, thank you. We deleted the extraneous word "and".

*Page 9*
*Caption for Figure 3 (c): … The exact STD of "Zerr decreases" as a increases*

Response: Yes, thank you, we have corrected this typo.

**Second Reviewer comments (in italics) and responses**

*The authors have addressed most of my comments to my satisfaction, but I still have some points left.*

(1) *In their response to calibration issues, the authors neglect scaling errors, i.e., the term a in $X_{cal} = aX+b$, with b the calibration bias. This may also be important, as was raised by Vogelzang in his supplement discussion. I recommend the authors to include this.*

Response: We considered the scaling errors in Appendix A3 the related paper AMT-2017-487 "Estimating observation and model error variance using multiple data sets." It is outside the scope of this paper to do a detailed comparison of the 3CH and TC methods using calibrated data according to this error equation and associated scaling coefficients a and b. However, we have modified the text in Section 5 of this paper (AMT-2018-75) by adding a sentence to line 15 (page 16): "..calibrating the data sets to the ERA-Interim

data set using the scaling coefficients as given by Stoffelen (1998) and Vogelzang et al. (2011)."

*(2) I recommend to omit paragraph 3.3 and move relevant information to the rest of the paragraph. If the authors wish, they can start paragraph 3 with a short outline.*

Response: We think the short summary is useful to readers and prefer to keep it. Putting the material in the rest of the paragraph would not save much space.

*(3) Page 1, line 3: I suggest "Both methods assume that the data sets are well intercalibrated and that the errors are uncorrelated"*

Response: The "well-calibrated" assumption is not required in general for the 3CH method. But it is true that both methods require that the unknown errors of the data sets are not correlated, so the sentence is OK as written. The part about calibration in the 2CH method is discussed later in the paper. Please see revision in the next response.

*(4) Page 1, line 5: the reference to Braun et al. (2001): I don't know the exact style requirements of AMT, but is a reference in the abstract allowed? I leave that to the editor.*

We have revised lines 3-6 in the abstract and taken out the reference to Braun (2011). It now reads: "Both methods have been used in previous studies to estimate the error variances associated with of number of physical and geophysical data sets. A key assumption in both methods is that the errors of the data sets are not correlated, although some studies have considered the effect of the partial correlation of representativeness errors in two or more of the data sets."

*(5) Page 1, line 17: "is" instead of "are" (subject is "estimating")*

Thank you, we made the correction.

*(6) Page 1, line 18: "is" instead of "are"*

Thank you, we made the correction.

*(7) Page 2, line 17: I suggest "estimate both errors and linear calibration coefficients of surface winds"*

We made the suggested change.

*(8) Line 32: "X,Y, and Z"*

We made this change.

*(9) Page 5, line 4 and Page 6, line 2: Both start with "We first generate". I recommend to change line 2 of page 6 to something like "Next we generate."*

We accept this suggestion and changed the beginning of the first sentence in Section 3.1 to "In the calculation of the correlated errors, we first generate random error profiles $X_{err}$, $Y_{err}$, and $Q_{err}$."

*(10) Page 19 line 3: I recommend "simulated data" instead of "a specified error model"*

[revised manuscript text omitted]

---

## Author Response (AR3)

**Response to Associate editor comments on AMT 2018-75 "Evaluating two methods of estimating error variances using simulated data sets with known errors"**

**Associate Editor comments in italics are given below, followed by our responses:**

*Comments to the Author:*
*The authors well considered the reviewer and editor suggestions.*

*A last minor comment is that Stoffelen (1998) elaborated on the necessity of error modeling before calibration and validation of geophysical data in his Appendix A and in section A2 particularly elaborates on the fundamental weakness of 2-way comparisons. This is closely related to your new section 4.3 and you may want to refer to it.*

Section 4.3 deals with the effect of error correlations, not biases nor calibration. Perhaps the editor means Section 4.4.3, which is a short section on the general effect of error biases. We do not feel that it is appropriate to include specific methods of dealing with biases in this section. In addition, we have already referred to Stoffelen's error modeling and calibration in Section 5 of this paper, and in more detail in Appendix A3 of Anthes and Rieckh (2017) (AMT-2017-487).